# Cooling low-dimensional electron systems into the microkelvin regime

Lev V. Levitin [1✉], Harriet van der Vliet[1,4], Terje Theisen [1], Stefanos Dimitriadis [1,5], Marijn Lucas[1], Antonio D. Corcoles [1,6], Ján Nyéki [1], Andrew J. Casey [1], Graham Creeth[2,7], Ian Farrer [3,8], David A. Ritchie [3], James T. Nicholls [1] & John Saunders[1]

Two-dimensional electron gases (2DEGs) with high mobility, engineered in semiconductor heterostructures host a variety of ordered phases arising from strong correlations, which emerge at sufficiently low temperatures. The 2DEG can be further controlled by surface gates to create quasi-one dimensional systems, with potential spintronic applications. Here we address the long-standing challenge of cooling such electrons to below 1 mK, potentially important for identification of topological phases and spin correlated states. The 2DEG device was immersed in liquid $^3$He, cooled by the nuclear adiabatic demagnetization of copper. The temperature of the 2D electrons was inferred from the electronic noise in a gold wire, connected to the 2DEG by a metallic ohmic contact. With effective screening and filtering, we demonstrate a temperature of 0.9 ± 0.1 mK, with scope for significant further improvement. This platform is a key technological step, paving the way to observing new quantum phenomena, and developing new generations of nanoelectronic devices exploiting correlated electron states.

[1] Department of Physics, Royal Holloway, University of London, Egham TW20 0EX, UK. [2] London Centre for Nanotechnology, University College London, London WC1H 0AH, UK. [3] Cavendish Laboratory, University of Cambridge, JJ Thomson Avenue, Cambridge CB3 0HE, UK. [4] Present address: Oxford Instruments Nanoscience, Abingdon, Oxfordshire OX13 5QX, UK. [5] Present address: Department of Physics, Imperial College London, London SW7 2AZ, UK. [6] Present address: Thomas J. Watson Research Center, Yorktown Heights, NY 10598, USA. [7] Present address: Praesto Consulting, Dublin D02 A342, Ireland. [8] Present address: Department of Electronic and Electrical Engineering, University of Sheffield, Sheffield S1 3JD, UK. ✉email: l.v.levitin@rhul.ac.uk

Two-dimensional electron gases (2DEGs), created at a GaAs–AlGaAs heterojunction and grown by molecular-beam epitaxy (MBE), have been the building block for the study of low-dimensional physics over the past few decades. This requires the high crystal perfection of MBE growth and the use of modulation doping (see ref. [1] for a review of the state of the art).

The lowest possible electron carrier temperature coupled to the highest sample quality is key to discovering and elucidating new ground states in such systems. The observation of the integer quantum Hall effect[2] and shortly thereafter the fractional quantum Hall effect (FQHE)[3] has led to extensive studies of two-dimensional electron systems in perpendicular magnetic field $B$. The lowest Landau level (LL), with filling factors $v < 1$ (here $v = nh/eB$ is determined by the carrier density $n_{2D}$, Planck's constant $h$ and elementary charge $e$) for the lowest spin branch is described by composite fermion (CF) theory[4]. It exhibits a rich variety of quantum states: fractional quantum Hall states for odd fractions, while $v = 1/2$ is a composite fermion Fermi sea[5]. A key question is the role of residual CF interactions in giving rise to exotic states[6]. A Wigner solid is found for small $v$, which also exhibits re-entrance with a maximum melting temperature of 50 mK above the $v = 1/3$ FQHE state[7]. Interactions are also believed to play a role at other fractional fillings such as $v = 4/11$, which has a measured activation energy of 5 mK[8].

The nature of quantum states in the second LL is in many cases not resolved, and this 2D electron system is of particular interest for study to lower temperatures than hitherto achieved. We discuss a few examples here. Even denominator states forming in the second Landau level are thought to arise from pairing of composite fermions. Confirmation of exact quantization of the 5/2 state first discovered by Willett et al.[9] required electron temperatures as low as 8 mK[10]. Nevertheless the topology of the $v = 5/2$ ground state (Pfaffian, anti-Pfaffian or other) remains an open question. Observations are highly sensitive to disorder[11,12]. It has been shown that a quantum phase transition to a quantum Hall nematic phase can be induced by applied pressure in both the 5/2 and 7/2 state[13]. The importance of such topological ground states is reinforced by the proposal that they offer a route to topological quantum computing using non-Abelian anyons[14]. A non-Abelian phase is also predicted for the 12/5 state.

Many of these states have small energy gaps. The 12/5 state has a gap of order 30 mK, which exhibits non-trivial tilt dependence of the magnetic field. Studies of the energy gaps at other fractional fillings[15,16] found that the gaps of the $v = 2 + 6/13$ (10 mK) and 2+2/5 (80 mK) states do not follow the CF hierarchy, leading to the proposal of a parton state[17]. The 2+3/8 state also has a reported gap of only 10 mK. Furthermore, there are indications of novel topological order in the upper spin branch of the second LL, at fillings 3+1/3 (37 mK) and 3+1/5 (104 mK)[18].

New states arising from strong correlations in the low-dimensional electron system are also expected in low magnetic fields. Electrostatic confinement of the 2DEG to form 1D wires and zero-dimensional (0D) quantum dots has led to studies of the many-body physics of Luttinger liquids[19,20] and the Kondo effect[21,22]. Furthermore the conduction electrons in GaAs-based 2DEGs are expected to couple to the nuclear spins, undergoing a collective ferromagnetic transition mediated by RKKY interactions which is expected at electron temperatures in the mK regime[23]. The hyperfine interaction is also predicted[24] to create new topological phases in a semiconductor 1D wire coupled to a superconductor and magnetic moments. A transition from metal- to insulator-like behaviour with decreasing temperature has been reported in a 2D hole gas in GaAs[25], potentially related to many-body localization. Recently, Wigner crystal-like zigzag chains have been discovered in 1D wires at low electron densities[26]. At lower temperatures there is potential for quantum communication

through 1D spin chains, coupling high fidelity quantum dot qubits[27].

Advances in cryogen-free technologies now make low milli-kelvin temperature platforms widely accessible[28], with prospects of commercial solutions in the microkelvin range[29,30]. However, cooling electrons in semiconductor devices is challenging. In situ cooling via the electron–phonon coupling is severely limited by its strong temperature dependence, while the electrons are susceptible to heat leaks arising from electromagnetic noise. Electromagnetic shielding and ultra-sensitive measurement schemes with low or no electrical excitation[31–33] are common strategies for minimising the heat leak. Cooling via the electrical leads has been widely implemented[10,33–36], commonly employing immersion in liquid helium. For 2D electrons, this has so far been limited, for many years, to 4 mK, see ref. [34] and references therein. Primary thermometers within samples confirmed 2D electron temperatures of 6 mK[33,36].

In response to the challenge to cool nano-electronic devices to ultra-low temperatures (ULT), new approaches have been introduced which focus on refrigeration by nuclear demagnetisation of each lead connected to the sample[37]. More recently this has been extended by incorporating nuclear refrigeration elements into the device itself. For this on-chip cooling the Coulomb blockade device used is itself a thermometer, providing an accurate measurement of the electron temperature in mesoscopic metallic islands as low as 0.5 mK[38,39].

In contrast, and complementary to this approach, the focus of our work is the cooling of a relatively large area (8 mm²) 2DEG. Our approach to cooling is motivated by the requirement for flexibility to cool a wide range of devices in different sample environments. We report an ultimate electron temperature in the 2DEG of 0.9 ± 0.1 mK, cooled in a ³He immersion cell, achieved after improvements in electromagnetic shielding. The 2DEG device is immersed in liquid ³He and cooled through metallic ohmic contacts, which are coupled to the ³He via compact heat exchangers made of sintered Ag powder. Cooling of devices by the immersion cell technique, as compared to on-chip cooling, is remarkably versatile due to modularity of the design. The cooling is provided by a physically remote copper nuclear adiabatic demagnetization stage module, so the magnetic field applied to the sample can be independently adjusted. A broad class of devices can potentially be cooled in this way, and in some cases the ³He liquid pressure can be used as an experimental control parameter[40].

The effective cooling of electrons in the 2DEG relies on both reducing the resistance (and hence thermal resistance via the Wiedemann-Franz law) of the 2DEG and ohmic contacts, and reducing the heat leak to the device. We fabricated a high-mobility GaAs-based 2DEG with sub-1 Ω AuNiGe ohmic contacts, see Fig. 1a, b, and reached a 1 fW level of heat leak to the 2DEG, see Fig. 1c, through designing a metal cell that is tight to photons over a wide frequency range, combined with extensive filtering of all electrical lines, see Fig. 2.

A key feature of the experiment was the choice of thermometer. We use a current sensing noise thermometer (NT) as an external module attached to the 2DEG via an ohmic contact. A SQUID current sensor is used to read out the voltage fluctuations across a gold wire[41]. Due to the fundamental Nyquist relation the NT operates over 5 orders of magnitude in temperature with a single point calibration. It can be configured as a primary thermometer[42], and as such underpins the recent redefinition of the Kelvin.

The noise thermometer dissipates no power in the device, but we must take account of both the inevitable residual heat leak into the thermometer, and the potential parallel cooling channel by direct coupling to the ³He in which it is immersed. We have

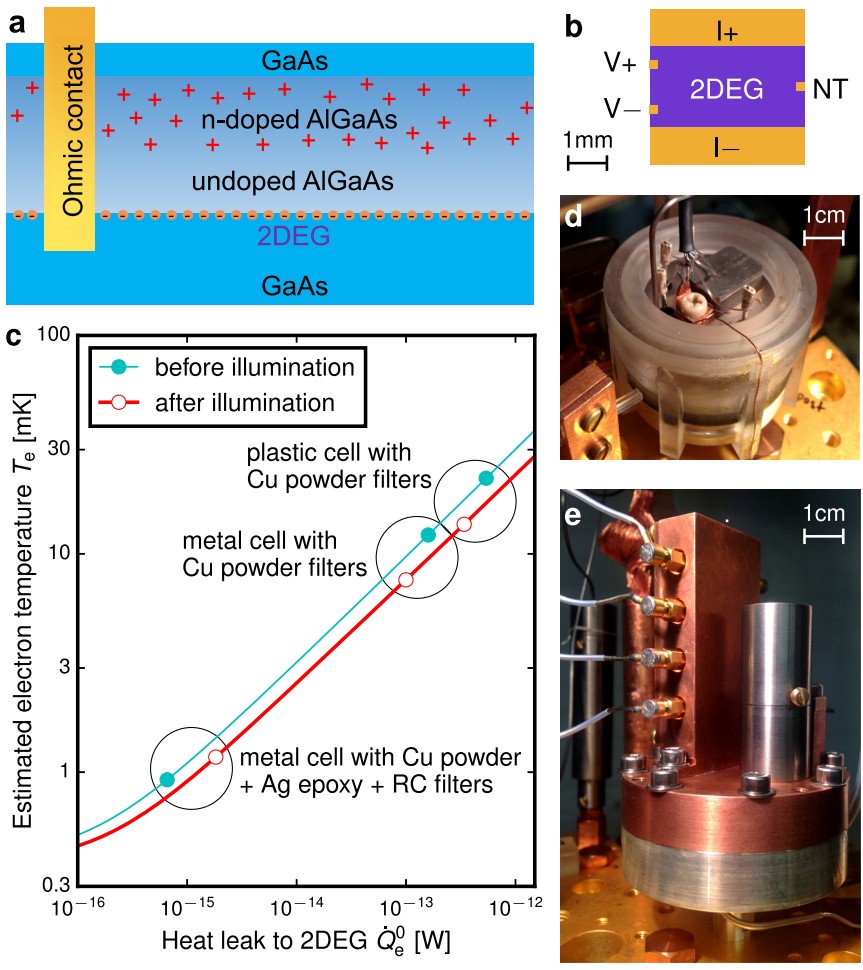

**Fig. 1 Cooling electrons in a 2DEG in $^3$He immersion cells. a** Schematic of the sample: the 2DEG is created at a GaAs/AlGaAs heterojunction, and electrical connections are provided by AuNiGe ohmic contacts. **b** Top view of the 4 mm × 4 mm device cooled to 1 mK by the I± and V± ohmic contacts, also used to measure the electrical resistances in the device. A noise thermometer (NT) is connected to the NT ohmic contact to probe the electron temperature $T_e$. **c** Progressive cooling of the 2DEG device in plastic (**d**) and metal (**e**) immersion cells equipped with different low-pass filters. $T_e$ is estimated from Eq. (4), assuming that the $^3$He bath is at temperature $T_{bath} = 0.3$ mK.

constructed a thermal model which demonstrates that at the lowest temperatures the NT accurately reflects the electron temperature in the 2DEG. The ability to alter the resistance of the 2DEG in situ by optical illumination, due to persistent photo-conductivity, was used to experimentally validate the model.

## Results

**Experimental setup**. Figure 2a is a schematic of the metal immersion cell, which combines a $^3$He bath with a photon-tight environment. Filtered connections to room-temperature electronics are provided to probe electrical resistances in the device and to operate heaters in thermal transport experiments. Two NTs are connected to the 2DEG and the demagnetisation refrigerator. Illumination by a red light-emitting diode (LED) facing the 2DEG allows us to increase the 2D carrier density and mobility[43]. Preliminary studies were conducted in a plastic cell shown in Fig. 1d, which was based on ref. [34]. See Supplementary Information (SI) for further details.

**Electrical transport measurements**. Figure 3a shows that the resistance of both the 2DEG and ohmic contacts exhibit a weak temperature dependence. These transport measurements were performed down to 1 mK with no discernible heating using conventional room temperature electronics without a screened

room; this demonstrates the effectiveness of the filtering. A change in the resistance was observed between 0.4 and 0.8 K, that has been identified in similar samples[44] to be a superconducting transition in the ohmic contacts.

**Thermal transport measurements and thermal model**. The use of an external thermometer in our setup requires an investigation of the heat flow in the system in order to relate the electron temperature in the 2DEG $T_e$ to the 2DEG NT temperature $T_{NT}$. We consider two sources of heat: $\dot{Q}_{NT} = \dot{Q}_{NT}^0 + \dot{Q}_{NT}^J$ applied to the NT and $\dot{Q}_e = \dot{Q}_e^0 + \dot{Q}_e^J$ uniformly generated within the 2DEG device. Each includes a residual heat leak ($\dot{Q}_{NT}^0$ and $\dot{Q}_e^0$); additionally, to study the thermal response Joule heating $\dot{Q}_{NT}^J$ is applied using the NT heater and $\dot{Q}_e^J$ by driving a current between I+ and I− ohmic contacts.

Figure 3b shows the thermal conductance $G_\Sigma(T_{NT}) = d\dot{Q}_{NT}/dT_{NT}$ between the 2DEG NT and the $^3$He bath, as inferred from measurements of $T_{NT}$ as a function of $\dot{Q}_{NT}$ at a constant $^3$He bath temperature $T_{bath}$. Figure 3c illustrates the two parallel channels of $G_\Sigma = G_e + G_{He}$, $G_e$ via electrons in the 2DEG and ohmic contacts, and the Kapitza boundary conductance $G_{He}$ between the NT assembly and the $^3$He bath[45]. The illumination of

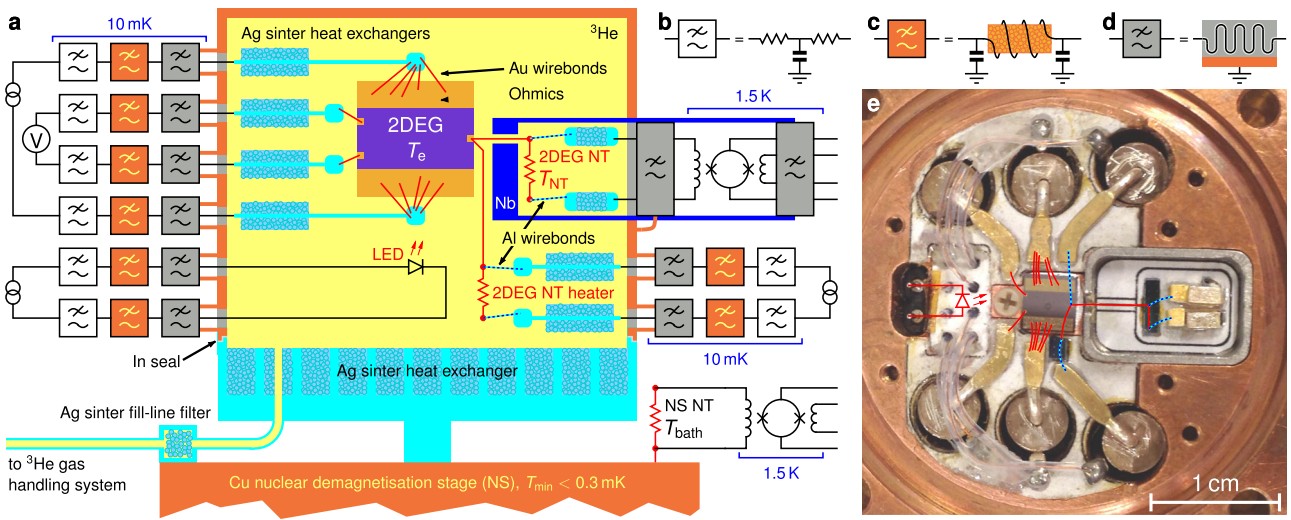

**Fig. 2 Experimental platform for cooling low-dimensional electrons. a** Schematic diagram of the metal immersion cell. Silver sinter on the silver base thermalises the liquid $^3$He to the nuclear demagnetisation refrigerator. The copper lid holds the 2DEG device, the sinter heat exchangers for cooling the I± and V± lines, the 2DEG noise thermometer (NT), and a red light-emitting diode (LED) for illuminating the 2DEG. The NT and its heater were connected using wedge-bonded resistive Au wires and superconducting Al wires acting as thermal breaks[62]. Another NT is mounted on the nuclear stage (NS) to measure the $^3$He bath temperature $T_{bath}$. The base and the lid sealed with In comprise a He-tight and photon-tight enclosure, complete with a Ag sinter plug in the filling capillary. All incoming electrical connections are fitted with low-pass filters of three types: **b** discrete $RC$ filters with $R = 1\,k\Omega$ and $C = 10\,nF$, **c** insulating Cu powder filters[63], and, **d** Ag epoxy filters with 1 m-long NbTi wires embedded in conducting Ag epoxy. RF-tight coaxial connections were used between these filters and the walls of the immersion cell. Ag epoxy filters with multiple wires, compatible with SQUID NT readout, were used on the SQUID lines. **e** The inside of the copper lid of the cell, with the Au and Al wires highlighted and the LED shown schematically. The NT is housed in a Nb shield, shown here with the top removed.

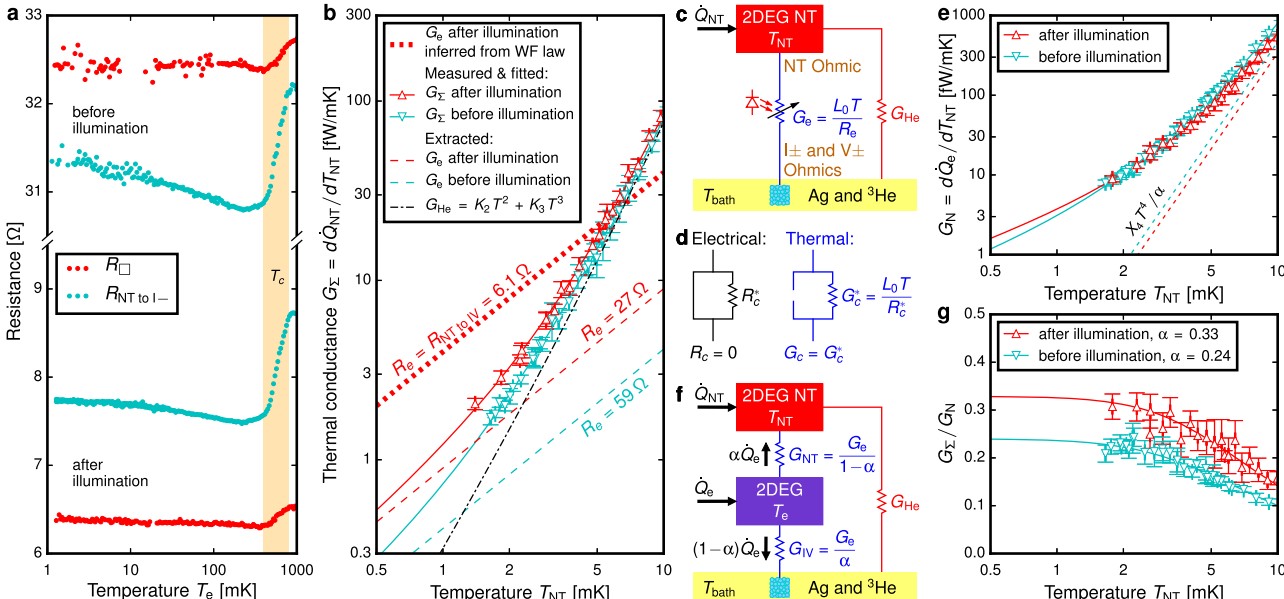

**Fig. 3 Transport measurements of 2DEG and ohmic contacts. a** Sheet resistance $R_{\square}$ of the 2DEG (red) and the resistance including 2DEG and two ohmic contacts (cyan) measured down to 1 mK. Weak temperature dependence of the resistance is observed in both the 2DEG and the ohmic contacts below the superconducting transition in the contacts which occurs at $T_c = 0.6 \pm 0.2$ K. The electron temperature $T_e$ is determined from Eq. (4). **b** Thermal conductance $G_\Sigma$ between 2DEG NT and $^3$He bath before (cyan) and after (red) illuminating the 2DEG, modelled as a sum of two parallel channels (**c**): $G_e$ via electrons in the 2DEG and ohmic contacts, sensitive to illumination, and $G_{He}$ due to immersion of the gold wires in liquid helium. Assuming the Wiedemann-Franz (WF) law applies in the 2DEG and ohmic contacts, after illumination the electron channel $G_e^{WF} = L_0 T / R_{NT\,to\,IV}$ alone exceeds the measured $G_\Sigma$. We attribute the violation of the WF law to superconductivity within the ohmic contacts. The effective electrical resistance $R_e$ is obtained by fitting $G_\Sigma(T)$ in (**b**) to Eq. (1). **d** In the partially-superconducting ohmic contact model the thermal conductance at $T \ll T_c$ is determined by the non-superconducting channel $R_c^*$ according to the WF law. **e** Response of the 2DEG NT to heating the 2DEG, the non-local thermal conductance $G_N$, is obtained analogously to $G_\Sigma$ and fitted to Eq. (2). Dashed lines show the $X_4 T^4$ term in Eq. (2), unimportant below $T_{NT} = 3$ mK. **f** A lumped-element thermal model described by Eqs. (3) and (4), in which $\dot{Q}_e$ is assumed to be applied in the centre of the 2DEG, splitting $G_e$ into $G_{NT}$ and $G_{IV}$, each including the 2DEG and the ohmic contacts. **g** The ratio $G_\Sigma/G_N$ describes the fraction of $\dot{Q}_e$ that flows into the NT. The $T \to 0$ limit $\alpha$ of $G_\Sigma/G_N$ governs $G_{NT}$ and $G_{IV}$ in (**f**). Error bars in **b**, **e**, **g** represent s.d.

the 2DEG increases $G_e$, allowing us to separate it from $G_{He}$. To reduce $G_{He}$ the heat exchangers in the immersion cell were plated with approx. 30 μmol/m² $^4$He coverage before loading $^3$He[46].

In the initial stage of this research we assumed that both the 2DEG and the ohmic contacts obey the Wiedemann-Franz (WF) law, ubiquitous in electronic transport in the $T \to 0$ limit[47–52]. The WF law predicts that $G_e^{WF} = L_0 T / R_{NT to IV}$, where $R_{NT to IV} = 27$ Ω (6 Ω) is the electrical resistance from the NT ohmic contact to I± and V± connected in parallel, measured before (after) illumination, where $L_0 = \pi^2 k_B^2 / 3e^2 = 2.44 \times 10^{-8}$ WΩ/K² is the Lorenz number[47]. Figure 3b shows that $G_\Sigma < G_e^{WF}$ after illumination, a clear violation of the WF law that we attribute to superconductivity[53] in the ohmic contacts[44]. To the leading order the change in $G_\Sigma$ due to illumination is $\Delta G_\Sigma = \Delta G_e \propto T$; to explain this observation we propose a model of partially-superconducting ohmic contacts, see Fig. 3d, such that well below the superconducting transition temperature $T_c$, $G_e = L_0 T / R_e$ is described by a resistance $R_e \geq R_{NT to IV}$. The data are well described by

$$G_\Sigma(T) = L_0 T / R_e + K_2 T^2 + K_3 T^3, \tag{1}$$

with $R_e = 59$ Ω (27 Ω) before (after) illumination and illumination-independent $G_{He} = K_2 T^2 + K_3 T^3$. Similarly the response to $\dot{Q}_e$ was characterised in terms of the non-local thermal conductance $G_N(T_{NT}) = d\dot{Q}_e / dT_{NT}$, see Fig. 3e, which was found to follow

$$G_N(T) = \left( G_\Sigma(T) + X_4 T^4 \right) / \alpha, \tag{2}$$

with an illumination-independent term $X_4$, but with different values of $\alpha$ before and after illumination. The $X_4 T^4$ term describes an additional cooling mechanism in the 2DEG or ohmic contacts, insignificant below $T_{NT} = 3$ mK, where $X_4 T^4 \ll G_\Sigma(T)$. In this low temperature regime, Eq. (2) reduces to $G_N = G_\Sigma / \alpha$ where $\alpha$ is the fraction of $\dot{Q}_e$ that flows towards the NT ohmic contact, as shown in Fig. 3f, g. Then

$$\dot{Q}_{NT} + \alpha \dot{Q}_e = \int_{T_{bath}}^{T_{NT}} G_\Sigma(T)\, dT = \frac{L_0}{2R_e} \left( T_{NT}^2 - T_{bath}^2 \right) \\ + \frac{K_2}{3} \left( T_{NT}^3 - T_{bath}^3 \right) + \frac{K_3}{4} \left( T_{NT}^4 - T_{bath}^4 \right). \tag{3}$$

The combined heat leak $\dot{Q}_{NT}^0 + \alpha \dot{Q}_e^0$ can be inferred from the difference between $T_{NT}$ and $T_{bath}$ in the absence of Joule heating. We assume $T_{bath}$ to be equal to the fridge temperature, as justified in SI. By replacing the gold thermal link between the NT ohmic contact and the NT with an aluminium one, we measured $\dot{Q}_{NT}^0 = 0.08 \pm 0.02$ fW in a separate experiment (see SI), allowing us to obtain $\dot{Q}_e$ directly, see Fig. 4a. Before illumination by the LED the heat leak was found to be $\dot{Q}_e^0 = 0.7 \pm 0.1$ fW. After illumination there was a dramatic increase of $\dot{Q}_e^0$, followed by a slow relaxation, consistent with slow recombination processes in the heterostructure[54]. On a time scale of a month, a new stable level $\dot{Q}_e^0 = 1.9 \pm 0.4$ fW was reached, higher than before the illumination.

To estimate the minimum electron temperature in the 2DEG we use the simple model shown in Fig. 3f, in which $\dot{Q}_e$ is a point source of heat in the middle of the device and additional cooling associated with the $X_4 T^4$ term in Eq. (2) is ignored:

$$T_e^2 = (1-\alpha) T_{bath}^2 + \alpha T_{NT}^2 + 2\alpha(1-\alpha) R_e \dot{Q}_e / L_0. \tag{4}$$

Figures 4b and c show the inferred electron temperature $T_e$, demonstrating that we have cooled the electrons to 1 mK, despite the unexpected hindrance posed by the superconductivity in the ohmic contacts.

## Discussion

We demonstrate that at 1 mK the dominant cooling of the 2D electrons is by the ohmic contacts, which have a thermal conductance of $G_e \sim 10^{-9}$ W/K² $\times T$. The electron–phonon coupling in high-mobility 2DEGs has been observed[55] to follow $\dot{Q}_{e-ph} = 61$ eV/s K⁵ $\times N(T_e^5 - T_{ph}^5)$ at phonon temperatures of $T_{ph} = 0.3-0.5$ K, where $N$ is the number of electrons. For our device this leads to a thermal conductance $G_{e-ph} = d\dot{Q}_{e-ph}/dT_e \sim 10^{-6}$ W/K⁵ $\times T^4$. Assuming this high-temperature power law extrapolates to ULT, the cross-over $G_e = G_{e-ph}$ occurs near 100 mK and at 1 mK $G_{e-ph}/G_e = 10^{-6}$, so cooling via electron–phonon coupling alone would only achieve $T_e = 20$ mK (70 mK) for $T_{ph} = 0.3$ mK and $\dot{Q}_e^0 \sim 1$ fW (500 fW).

The $X_4 T^4$ term in Eq. (2) points towards an additional cooling channel competing with $G_e$ above 3 mK, such as enhanced $G_{e-ph} \gg 10^{-6}$ W/K⁵ $\times T^4$ at ULT; alternatively $G_e(T)$, $G_{NT}(T)$ and $G_{IV}(T)$, see Fig. 3f, may deviate from the $G \propto T$ behaviour, if the effective contact resistance $R_c^*$ (see Fig. 3d) is not constant at ULT. Ignoring these effects in Eq. (4) potentially overestimates $T_e$ for the preliminary experiments shown in Fig. 1c, but the ultimate electron temperature we report and the qualitative observation of the reduction of the heat leak $\dot{Q}_e$ due to the electromagnetic shielding are robust. In addition to the $X_4 T^4$ term, a more detailed thermal model, beyond the scope of this work, should take into account the distribution of $\dot{Q}_e$ across the 2DEG and 2D and 3D heat flow in the 2DEG and ohmic contacts.

We note that after illumination the thermal resistances are dominated by the ohmic contacts, since $R_e \gg R_{NT to IV}$. Therefore the thermal conductance through the I± and V± contacts in parallel, $G_{IV} = G_e/\alpha$, (see Fig. 3f), is only $(1-\alpha)/\alpha = 2.1$ times higher than that through the NT ohmic, $G_{NT} = G_e/(1-\alpha)$, despite being 15 times larger in circumference. This suggests that the large I± ohmics were rendered thermally inactive due to low thermal conductance along them, as the gold wires were bonded 0.5 mm away from their front edges, Fig. 2e, to prevent damage to the 2DEG adjacent to these ohmics. The larger value of $(1-\alpha)/\alpha = 3.2$ before illumination is consistent with the I± ohmics being more active when the 2DEG resistance is higher.

A natural question to ask is whether even lower electron temperatures can be achieved. First, the heat leak may be reduced by optimising the device geometry (for some sources $\dot{Q}_e^0 \propto$ area ) and further filtering. We recognise that our NT readout scheme limited the filters between the NT and the SQUID to a two-way Ag epoxy filter with a high cut-off frequency. Improved electron cooling is expected in experiments compatible with heavy filtering of every measurement line.

Another approach is to improve the ohmic contacts. Their thermal conductance can be increased if the superconductivity is suppressed with a magnetic field or by a change of recipe. To optimise the performance of the large partially-superconducting ohmic contacts used in this work, a thick gold film could be evaporated on the annealed top surface or a dense network of gold wires could be bonded along the front edge of the contacts. We estimate electron temperatures of 0.4–0.6 mK, if the above steps are combined with the optimised fridge performance (see SI).

Important future steps include characterising and mitigating the heating associated with surface gates used to define mesoscopic samples, measuring the electron temperature in the device directly[33,56], and extending the techniques presented here to high magnetic fields. Based on the thermal measurements at 6–100 mK[33], the quantum Hall states will cool into the microkelvin regime in our environment. The immersion cooling can efficiently thermalise a variety of degrees of freedom in condensed matter systems, offering a promising path to control the decoherence problem in superconducting electronics[57].

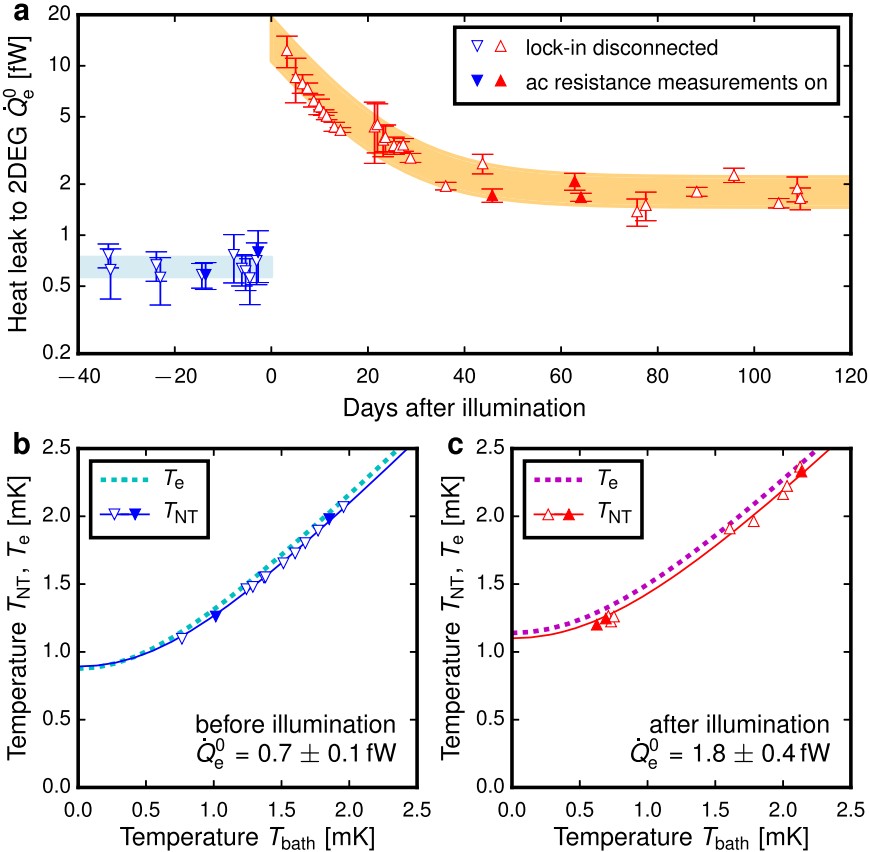

**Fig. 4 Heat leak to the electrons in 2DEG and inferred electron temperature before (blue) and after (red) illumination. a** The heat leak $\dot{Q}_e^0$ to the 2DEG, determined from Eq. (3). Error bars and thick lines through the data represent s.d. Low-excitation AC transport measurements (filled triangles) do not increase the heat leak significantly over the value it takes when no room-temperature electronics are connected to the experiment (open triangles). After illumination with the LED the heat leak increases dramatically and then gradually decays, stabilising a month later; the orange band that highlights the exponential decay is a guide to the eye. **b, c** The triangles represent the measurements of $T_{NT}$ vs $T_{bath}$ corresponding to the $\dot{Q}_e^0$ shown in (**a**); the solid lines are fits to Eq. (3). The dotted lines show the electron temperature $T_e$ in the 2DEG estimated from Eq. (4). The first 40 days after illumination are excluded from (**c**) due to evolving $\dot{Q}_e^0$.

In conclusion, the demonstrated cooling of a large area two-dimensional electron system to below 1 mK constitutes a significant technical breakthrough in quantum nanoelectronics. Fundamental studies of lower dimensional electron devices such as quantum wires and quantum dots in the microkelvin regime can be achieved if the heat leak to the 2DEG, generated by the surface gates, is kept at the fW level. Likewise, the addition of a strong magnetic field, which is straightforward to implement in our modular design, coupled with samples of the highest quality, is poised to contribute to the understanding of novel ground states of two-dimensional electron systems at fractional LL fillings, arising from CF interactions. Thus the microkelvin regime is opened up for the study of a rich and diverse array of strongly correlated quantum systems, with high potential for future discovery.

## Methods

**2DEG sample.** The 4 mm × 4 mm sample shown in Fig. 1b was fabricated using wafer W476; details of its MBE growth and the subsequent fabrication of AuNiGe ohmic contacts are given elsewhere[44], together with the critical field measurements of the superconducting state below 1 K. The as-grown 2D carrier density and mobility are $n_{2D} \approx 2 \times 10^{11}$ cm$^{-2}$ and $\mu \approx 1 \times 10^6$ cm$^2$/Vs giving a calculated sheet resistance of $R_\square \approx 31\,\Omega$. After illumination these quantities are $3.3 \times 10^{11}$ cm$^{-2}$, $3 \times 10^6$ cm$^2$/Vs and $6\,\Omega$. Illumination was performed at 1.5 K with the immersion cell evacuated. The LED was driven with currents up to 2 mA for 1 min, by which time $R_\square$ saturated.

**Filters.** The Ag epoxy filters, shown in Fig. 2d, combine features of several designs[32,58,59]. Metre-long superconducting NbTi wires were coiled around

threaded sterling silver tubes and encapsulated in conducting Ag epoxy. In the resulting lossy coaxial the dissipation occurs in the outer conductor which is in direct metallic contact with the refrigerator, ensuring good thermalisation. These filters with a 100 MHz cut-off were combined with the Cu powder filters with 2 MHz cut-off, and the 16 kHz low-pass $RC$ filters shown in Fig. 2b. To prevent high-frequency leaks the Ag epoxy filters were connected to the immersion cell via semi-rigid coaxial cables with threaded connectors.

A version of the Ag epoxy filter using a NbTi twisted pair (instead of a single wire) embedded in Ag epoxy was inserted between 2DEG NT and its SQUID sensor. This design was chosen for having zero DC resistance and for inducing sufficiently low amounts of noise in the SQUID so as to not mask the NT signal. This filter was mounted directly on top of the immersion cell and sealed to the cell wall with In to maintain a photon-tight enclosure. Another multi-line Ag epoxy filter at 1.5 K was inserted between the SQUID sensor and its room temperature electronics. All remaining filters were mounted at the mixing chamber plate of the dilution refrigerator.

**Noise thermometry.** Both NTs were read out with integrated 2-stage SQUID current sensors[60] and were calibrated against a primary magnetic field fluctuation thermometer at 10–100 mK[61].

**Electrical transport measurements.** were performed using a mains-powered Stanford Research Systems SR-124 analogue lock-in amplifier using its internal oscillator. Below 20 mK measurement currents between 1 and 10 nA were used to ensure sub-fW Joule heating.

**Thermal measurements.** The thermal conductance $G_\Sigma(T_{NT}) = d\dot{Q}_{NT}/dT_{NT}$ was inferred from pairs of measurements of $T_{NT} = T_{a,b}$ at different levels of $\dot{Q}_{NT}^J = \dot{Q}_{a,b}$ at constant $T_{bath}$ as $G_\Sigma((T_a + T_b)/2) = (\dot{Q}_b - \dot{Q}_a)/(T_b - T_a)$, Fig. 5a. Identical techniques were used for measuring $G_N(T)$. Below 3 mK the 2DEG NT responded slowly, Fig. 5b, and $T_{bath}$ could not be kept constant sufficiently long due to the

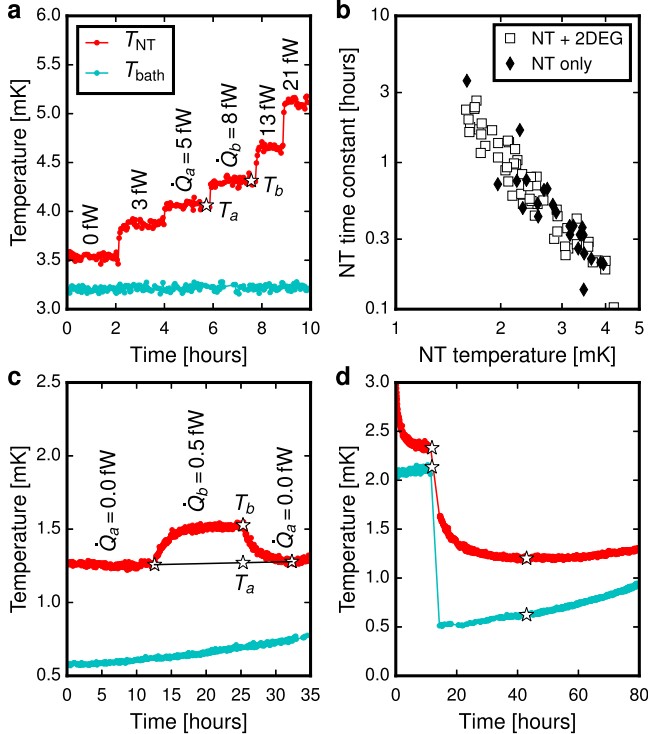

**Fig. 5 Measurements of the thermal conductance and heat leak. a** The thermal conductance $G_\Sigma$ was inferred from the step-wise measurement of $T_{NT}(\dot{Q}_{NT}^J)$ when $T_{bath}$ was kept constant. **b** The slow response of $T_{NT}$ to heaters and $T_{bath}$ below 3 mK required a modified $G_\Sigma$ measurement shown in (**c**). **d** Heat leak was inferred via stationary Eq. (3) which applies when $dT_{NT}/dt = 0$.

single-shot operation of the demagnetisation cooling, so $T_a$ was inferred from measurements before and after $T_b$, see Fig. 5c. Since similar slow relaxation was observed when NT was isolated from the 2DEG, see Fig. 5b, we conclude that its origin is the large heat capacity of the NT itself, and all other elements of the system thermalise faster. This implies that when $dT_{NT}/dt = 0$, see Fig. 5d, the entire system is in a steady state, allowing the stationary Eq. (3) to be used to extract the heat leak. The slow NT response practically limits the thermal measurements to $T_{NT} > 1.2$ mK.

In the early experiments shown in Fig. 1c, $T_{NT}$ was too high to ignore the $X_4 T^4$ term in $G_N$. Here the thermal model given by Eqs. (3) and (4) is inaccurate. The heat leak was inferred from $\dot{Q}_e^0 = \int_{T_{bath}}^{T_{NT}} G_N(T)\,dT$. To account for the use of different 2DEG devices of the same design and the lack of $^4$He plating $G_N(T)$ was measured separately in each experiment, see SI for further information.

## Data availability

The thermal conductance, electrical resistance, heat leak and thermal time constant data obtained in this work are available in Figshare https://doi.org/10.6084/m9.figshare.17057063.

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

## Acknowledgements

We thank P. Bamford, R. Elsom, I. Higgs and J. Taylor for mechanical support, and V. Antonov for wire bonding. Cu powder filters designed and fabricated by A. Iagallo and M. Venti were instrumental in the early stages of this work. We acknowledge fruitful discussions with C. Ford, R. Haley, P. Meeson, P. See and J. Waldie. This research was supported by EPSRC Programme grant EP/K004077/1 and the EU H2020 European Microkelvin Platform EMP, Grant No. 824109.

## Author contributions

J.S., A.D.C., J.N., A.J.C. and L.V.L. conceived the experiments; the wafer was grown by D.A.R. and I.F.; A.D.C., G.C. and J.T.N. designed and fabricated the 2DEG device; A.D.C. and L.V.L. designed and constructed the plastic immersion cell, with input from J.N., A.J.C., J.T.N. and J.S. L.V.L. and H.v.d.V. designed and constructed the metal immersion cell and filters and performed ultra-low temperature measurements in both cells, with input from J.N., A.J.C., J.T.N. and J.S.; S.D. and L.V.L. measured the heat leak to the 2DEG; T.T., M.L., A.J.C. and J.T.N. characterised the superconductivity in the ohmic contacts; L.V.L. analysed the data; L.V.L., J.S. and J.T.N. wrote the manuscript with critical input from all authors.

## Competing interests

The authors declare no competing interests.
