## [Peer Review File · Nature Communications]

Cooling Low-Dimensional Electron Systems into the Microkelvin RegimeREVIEWERS' COMMENTS

Reviewer #1 (Remarks to the Author):

The manuscript reports on the achievement of an electronic temperature of 1mK in a 2D electron gas. Whereas electronic temperatures below 1mK were previously demonstrated in small circuits, these concerned only extremely specific devices incorporating in themselves a cooling mechanism (and a thermometer). The present demonstration does not involve such an in-situ cooling, but was achieved on a widely used 2D electron gas (2DEG) where the previously lowest electron temperature was of about 4mK. It therefore shows a path to obtain ~ 1 mK electronic temperatures for a possibly large class of nanoscale circuits.

In practice, this work does not develop novel cooling, thermalisation or thermometry strategies. The observed low temperature improvement mostly results from the elimination of external heat leaks through a very thorough filtering of the connected lines and the elimination of most plastics in the sample vicinity. A particular challenge is to determine reliably the temperature of electrons in the 2DEG. In this work, this temperature is not directly measured, but inferred based on a thermal model from the noise temperature of a close by electrically connected system. Although less direct, the authors took great care to ascertain the reliability of their thermometry (an interesting study in itself). It does not seem possible that any uncertainty could erase the claimed great improvement over previous lowest 2DEG temperatures.

The manuscript is well written, both accessible and providing the necessary details on the system with an adequate distribution between main text, methods and supplementary.

Overall, I personally believe that this manuscript fully qualifies to the milestone and broad interest criteria of Nature communications, given the remarkable technical step that constitutes a factor four improvement on the temperature in 2DEGs and given the widespread importance of temperature. I recommend publication without mandatory modifications.

Nevertheless, there are a few points that could be addressed to improve the paper:

- An additional X4T4 term is required to fit the data above ~ 3 mK. Its impact can be seen in the non-constant value of G_{sigma}/G_N shown Fig. 3g, yet it would be easier to appreciate its effect if the corresponding contribution (included in the continuous lines in Fig. 3e if I understood correctly) would also be separately displayed, eg as dashed lines in Fig. 3e. Although its origin is unknown, the authors could propose what they believe are the most likely possibilities for this extra mechanism.
- The authors mention L325 that the heat leak Q_{0e} is proportional to the area. Is that so clear? It seems, for example, that this would not be true of the contribution from unfiltered radiations from the connected lines.
- As I understood it, T_{bath} is not directly measured but a fit parameter, but I did not find this information directly and clearly stated. Could the authors explicitly, and early on clarify this point? If it turns out it is measured, a bit more details on how this is done should be provided.
- Given the practical challenge to reduce heat leakage at low temperatures, it would be interesting to find in the SI more information on the metal cell, such as the specific choices of insulating materials appearing white in Fig 2e, the "glue" used to stick the sample (silver paint or epoxy?)...

Reviewer #2 (Remarks to the Author):

Authors present a platform to cool electronic materials below 1mK. In order to achieve this, they used an immersion cell technology paired with electronic filtering techniques. A SQUID-based noise thermometer connected to a low impedance wire was used to infer on the temperature in the sample.

There is currently an active effort worldwide to cool electrons into the microkelvin temperature range. These efforts are motivated by a quest for new topological phases and will certainly influence various quantum technologies currently under development. Authors adapted an existing immersion cell technology to reach the microkelvin range. In all, the paper has reached several milestones in cooling and the combination of techniques used performs better than other existing technologies:

1. The B-field at the sample can be independently adjusted, and can be of zero value inclusively. This contrast, other techniques which use a nuclear refrigerant attached to the sample contacts. In these other setups measurements under zero field are not possible, as dropping the field all the way to zero will cause a heating of the sample.
2. The importance of rf filtering has long been known, these techniques were adopted in the present setup. However, in addition to filtering, authors convincingly demonstrate the importance of rf shielding. This is studied by comparing a plastic cell with a metal cell. Furthermore, shielding is also used on wires - other publications either do not use such shielding or do not discuss this. This shielding reduces a great deal the heatleak due to photons and thus enable reaching lower temperatures.
3. Authors recognize the importance of superconducting contacts and its effects on cooling electron in the sample. Superconductivity in the material system under study may not be avoided. However, mitigating factors are discussed.
4. The model for heatflow is very carefully developed, will be useful for the colleagues in the field. The manipulation of the thermal properties of the sample is novel.

To conclude, I support publication.

I ask the authors to consider adding a few details.

1. Cryogen-free platforms are mentioned in the intro, but I think authors do not explicitly state whether they used one.
2. What was the excitation level used for resistance measurements, such as the data from Fig.3 ? I am sure the level was low enough to avoid self-heating effects, but having that number would be nice.

Responses to the Referees' Comments

Reviewer #1 wrote:

The manuscript reports on the achievement of an electronic temperature of 1mK in a 2D electron gas. Whereas electronic temperatures below 1mK were previously demonstrated in small circuits, these concerned only extremely specific devices incorporating in themselves a cooling mechanism (and a thermometer). The present demonstration does not involve such an in-situ cooling, but was achieved on a widely used 2D electron gas (2DEG) where the previously lowest electron temperature was of about 4mK. It therefore shows a path to obtain "1mK electronic temperatures for a possibly large class of nanoscale circuits.

In practice, this work does not develop novel cooling, thermalisation or thermometry strategies. The observed low temperature improvement mostly results from the elimination of external heat leaks through a very thorough filtering of the connected lines and the elimination of most plastics in the sample vicinity. A particular challenge is to determine reliably the temperature of electrons in the 2DEG. In this work, this temperature is not directly measured, but inferred based on a thermal model from the noise temperature of a close by electrically connected system. Although less direct, the authors took great care to ascertain the reliability of their thermometry (an interesting study in itself). It does not seem possible that any uncertainty could erase the claimed great improvement over previous lowest 2DEG temperatures.

The manuscript is well written, both accessible and providing the necessary details on the system with an adequate distribution between main text, methods and supplementary.

Overall, I personally believe that this manuscript fully qualifies to the milestone and broad interest criteria of Nature communications, given the remarkable technical step that constitutes a factor four improvement on the temperature in 2DEGs and given the widespread importance of temperature. I recommend publication without mandatory modifications.

We cordially thank the reviewer for recognising the importance of our work and recommending the publication. We are glad the manuscript appears accessible and complete to the reviewer.

Nevertheless, there are a few points that could be addressed to improve the paper:

- An additional X_4T^4 term is required to fit the data above "3mK. Its impact can be seen in the non-constant value of G_{sigma}/GN shown Fig. 3g, yet it would be easier to appreciate its effect if the corresponding contribution (included in the continuous lines in Fig. 3e if I understood correctly) would also be separately displayed, eg as dashed lines in Fig. 3e. Although its origin is unknown, the authors could propose what they believe are the most likely possibilities for this extra mechanism.

We now illustrate the X_4T^4 term in Fig. 3e, as suggested. In the Discussion section we suggest that either an unknown parallel cooling mechanism may be at play, such as enhanced electron-phonon interactions, or the effective contact resistances R_c^* may have a

temperature dependence well below $T_c = 0.6 \pm 0.2$ K, possibly related to inhomogeneous superconductivity in the ohmic contacts. More experimental work is required to fully understand this behaviour, therefore we refrain from further speculations.

- The authors mention L325 that the heat leak Q_{0e} is proportional to the area. Is that so clear? It seems, for example, that this would not be true of the contribution from unfiltered radiations from the connected lines.

We agree with the reviewer that not all sources of heat scale with the device area. The sentence has been altered to make this clear.

- As I understood it, T_{bath} is not directly measured but a fit parameter, but I did not find this information directly and clearly stated. Could the authors explicitly, and early on clarify this point? If it turns out it is measured, a bit more details on how this is done should be provided.

We used the nuclear stage noise thermometer (NS NT) to measure T_{bath} , as stated in Fig. 2 caption. We now clarify this in text below Eq. (3), where the exact value of T_{bath} is first important, with a reference to the SI. A new SI section “ ^3He Temperature” demonstrates excellent thermalisation of ^3He in the immersion cell to the nuclear stage, measured directly in the plastic immersion cell and argues that this result holds for the metal cell. We are grateful to the reviewer for this question, as it urged data re-analysis, which strengthened the argument that the temperature gradient between the cooling platform and the ^3He bath is insignificant. The “2DEG noise thermometer correction” section in the SI has been updated accordingly.

- Given the practical challenge to reduce heat leakage at low temperatures, it would be interesting to find in the SI more information on the metal cell, such as the specific choices of insulating materials appearing white in Fig 2e, the “glue” used to stick the sample (silver paint or epoxy?)...

The “Further details of the experimental setup” section was created in the SI, which lists the materials used in construction of the experimental cells.

Reviewer #2 wrote:

Authors present a platform to cool electronic materials below 1mK. In order to achieve this, they used an immersion cell technology paired with electronic filtering techniques. A SQUID-based noise thermometer connected to a low impedance wire was used to infer on the temperature in the sample.

There is currently an active effort worldwide to cool electrons into the microkelvin temperature range. These efforts are motivated by a quest for new topological phases and will certainly influence various quantum technologies currently under development. Authors adapted an existing immersion cell technology to reach the microkelvin range. In all, the paper has reached several milestones in cooling and the combination of techniques used performs better than other existing technologies:

1. The B-field at the sample can be independently adjusted, and can be of zero value inclusively. This contrast, other techniques which use a nuclear refrigerant attached to the sample contacts. In these other setups measurements under zero field are not possible, as dropping the field all the way to zero will cause a heating of the sample.

2. The importance of rf filtering has long been known, these techniques were adopted in the present setup. However, in addition to filtering, authors convincingly demonstrate the importance of rf shielding. This is studied by comparing a plastic cell with a metal cell. Furthermore, shielding is also used on wires - other publications either do not use such shielding or do not discuss this. This shielding reduces a great deal the heatleak due to photons and thus enable reaching lower temperatures.

3. Authors recognize the importance of superconducting contacts and its effects on cooling electron in the sample. Superconductivity in the material system under study may not be avoided. However, mitigating factors are discussed.

4. The model for heatflow is very carefully developed, will be useful for the colleagues in the field. The manipulation of the thermal properties of the sample is novel.

To conclude, I support publication.

We are grateful to the reviewer for recognising the merits of our work and for recommending the publication.

I ask the authors to consider adding a few details.

1. Cryogen-free platforms are mentioned in the intro, but I think authors do not explicitly state whether they used one.

We now clarify that we did not use a cryogen-free platform, see “Further details of the experimental setup” section in the SI, created in response to Referee #1.

2. What was the excitation level used for resistance measurements, such as the data from Fig.3 ? I am sure the level was low enough to avoid self-heating effects, but having that number would be nice.

We now state the range of excitation current levels used at ultra-low temperatures in the “Electrical transport measurements” section of Methods.

We are grateful to both referees for the helpful and constructive requests that have improved the manuscript.

Changes to comply with the Author Checklist

- We added “Introduction” section heading.
- We explained “NT” abbreviation in Fig. 2 caption.
- We explained “ULT” abbreviation in text at page 2.
- We modified Fig. 3 caption to explain the meaning of all lines in Fig. 3b. We explain the meaning of the error bars in Figs. 3, S2, S5 and S6.
- We created a public data depository and provided a link to it in the main text.
- We reordered References, Data availability statement and Acknowledgements.

Other Changes

- We corrected units in Table S3 ($W/K^{4.2}$ to $W/K^{4.2}$, etc.).
- We changed “Thermal conductance” caption in “Methods” to “Thermal measurements”, since this section describes both thermal conductance and heat leak measurements.
- We changed “LED illumination” to “illumination with the LED” in Fig. 4 caption for clarity.
- We changed the order of SI sections to improve the page layout.
- We renamed “Noise thermometer correction” section in the SI to “2DEG noise thermometer correction”, to clarify that this section does not deal with the difference between the temperatures of nuclear stage noise thermometer and ^3He bath.